# LUMINA: Foundation Models for Topology Transferable ACOPF

**Yijiang Li**
Argonne National Laboratory
Lemont, IL 60439, USA
`yijiang.li@anl.gov`

**Zeeshan Memon**
Emory University
Atlanta, GA 30322, USA
`zeeshan.memon@emory.edu`

**Hongwei Jin**
Argonne National Laboratory
Lemont, IL 60439, USA
`jinh@@anl.gov`

**Stefano Fenu**
Argonne National Laboratory
Lemont, IL 60439, USA
`sfenu@@anl.gov`

**Keunju Song**
Sogang University
Seoul, South Korea
`kjsong4089@sogang.ac.kr`

**Sunash B Sharma**
Argonne National Laboratory
Lemont, IL 60439, USA
`sunash.sharma@anl.gov`

**Parfait Gasana**
Argonne National Laboratory
Lemont, IL 60439, USA
`pgasana@anl.gov`

**Hongseok Kim**
Sogang University
Seoul, South Korea
`hongseok@sogang.ac.kr`

**Liang Zhao**
Emory University
Atlanta, GA 30322, USA
`liang.zhao@emory.edu`

**Kibaek Kim**
Argonne National Laboratory
Lemont, IL 60439, USA
`kimk@anl.gov`

## Abstract

Foundation models in general promise to accelerate scientific computation by learning reusable representations across problem instances, yet constrained scientific systems, where predictions must satisfy physical laws and safety limits, pose unique challenges that stress conventional training paradigms. We derive design principles for constrained scientific foundation models through systematic investigation of AC optimal power flow (ACOPF), a representative optimization problem in power grid operations where power balance equations and operational constraints are non-negotiable. Through controlled experiments spanning architectures, training objectives, and system diversity, we extract three empirically grounded principles governing scientific foundation model design. These principles characterize three design trade-offs: learning physics-invariant representations while respecting system-specific constraints, optimizing accuracy while ensuring constraint satisfaction, and ensuring reliability in high-impact operating regimes. We present the LUMINA framework, including data processing and training pipelines to support reproducible research on physics-informed, feasibility-aware foundation models across scientific applications.

## 1 Introduction

Foundation models promise to accelerate scientific computation by learning generalizable representations from large-scale pretraining, enabling fast inference on new scenarios without expensive iterative solvers (Mao et al., 2024; Drakulic et al., 2024). However, many scientific and engineering

domains impose hard constraints, conservation laws, safety limits, or thermodynamic bounds that standard supervised learning often violates under distribution shift. Additionally, three key factors compound this challenge for scientific foundation models. First, models must maintain feasibility not only under nominal conditions seen during training but also under operational extremes where constraint margins tighten and feasibility boundaries become critical (Yuan et al., 2025). Second, scientific systems exhibit structural heterogeneity that foundation models must navigate, yet physical laws manifest differently in each system's structure (Yuan et al., 2022; Ódor & Hartmann, 2018). Third, practitioners require not just low mean prediction error but guaranteed feasible solutions and interpretable failure modes, since even rare infeasible predictions can render a surrogate unusable in critical workflows.

We use the term foundation model in the domain-specific sense that has emerged in the scientific ML literature: a model pretrained across diverse operating conditions and multiple system configurations that supports zero-shot transfer and fast adaptation to unseen instances. This is analogous to how the term has been applied in molecular and weather modeling contexts, rather than implying the billions-of-parameter, internet-scale pretraining of large language models (Shi et al., 2024).

In this paper, we use AC optimal power flow (ACOPF) (Carpentier, 1962; Momoh et al., 1999; Frank et al., 2012a;b) to study how foundation models can transfer across grid topologies while maintaining feasibility under hard physical constraints. Modern grid operations require millions of contingency evaluations for reliability assessment and planning, forcing ACOPF to be solved repeatedly at prohibitive computational cost on large networks. This has motivated a growing line of work on learning-based surrogates and foundation-style models for accelerated optimization (e.g., Pan et al., 2023a; Huang et al., 2022; Pan et al., 2023b; Zeng et al., 2024; Fioretto et al., 2020; Hien et al., 2025; Hamann et al., 2024; Stiasny & Cremer, 2026; Liu et al., 2025). The problem enforces nonlinear AC power flow equations as equality constraints and operational limits on generation, voltage, and line flows as inequality constraints, testing a model's ability to learn and respect physical laws. Power networks span diverse topologies from small distribution feeders to large transmission systems, providing controlled variation in structure while preserving common underlying physics. Unlike many scientific domains where ground truth is scarce, ACOPF benefits from mature solvers that provide high-quality labeled data across operating points and topologies, enabling rigorous empirical evaluation (Pirnia et al., 2013). While prior work demonstrates substantial speedups over iterative solvers in favorable settings, open questions remain regarding generalization across topologies, scaling to larger systems, and reliable constraint satisfaction under system shift.

Specifically, we conduct a comprehensive study of ACOPF surrogate learning across architectures, training regimes, loss function design, and evaluation scenarios, and extract empirically grounded principles for constrained scientific foundation models. In LUMINA (**L**arge-scale **U**nified **M**odel for **IN**telligent grid **A**pplications), we compare homogeneous and heterogeneous graph neural networks to characterize how explicit encoding of system structure affects generalization to unseen topologies and scaling to larger systems. We contrast standard supervised learning with Lagrangian-based augmented objectives that explicitly penalize constraint violations during training, measuring accuracy-feasibility trade-offs across single-topology, multi-topology, and zero-shot inference and finetuning scenarios. We quantify sample complexity for convergence on large systems and compare zero-shot transfer via fine-tuning against training from scratch. We provide diagnostic analyses on error attribution by topological properties, and failure mode characterization under high-load regimes to expose brittleness patterns and guide future improvements.

**Contributions.** 1) We introduce LUMINA, an open-source foundation model framework for ACOPF that systematically studies constrained scientific foundation model design. 2) We provide a controlled experimental analysis of how model architecture and training strategy design affect generalization and constraint satisfaction in nonlinear scientific systems. 3) We characterize scaling behavior and reliability under topology and load shift, providing practical guidance for deploying learned surrogates in constrained settings.

## 2 SCIENTIFIC QUESTIONS AND WORKFLOW CONTEXT

Our empirical investigations of ACOPF surrogate learning are motivated by how surrogates enter scientific workflows and what reliability guarantees practitioners require. In grid operations, ACOPF surrogates enable scenario screening across thousands of potential system states, what-if analysis

for planning under uncertainty, sensitivity studies that explore parameter spaces, and near-real-time planning loops where millisecond inference replaces minute-scale optimization. Each application imposes different reliability requirements, but all share a common constraint: the surrogate must produce physically feasible solutions, not merely accurate predictions in aggregate. A model that achieves strong average accuracy but occasionally violates power balance or thermal limits cannot be deployed in operational settings where such violations trigger protective equipment, cascade into broader failures, or force operators to reject the solution entirely and fall back to slower conventional methods. We therefore organize our study around three scientific hypotheses that link architectural and algorithmic choices to operational reliability:

**System Generalization**   The system generalization hypothesis tests whether multi-topology pre-training yields transferable representations that preserve feasibility across network structures, enabling zero-shot or few-shot adaptation to unseen grids. This examines whether foundation models internalize topology-agnostic physical principles, or whether structural variation necessitates specialized training for each network.

**Feasibility and Reliability**   The feasibility hypothesis evaluates whether constraint-aware training objectives reduce violation rates under distribution shift relative to purely supervised learning. Standard supervision optimizes predictive accuracy on labeled solutions but treats constraints as implicit correlations; we test whether explicit constraint penalization improves out-of-distribution feasibility.

**Hard-Regime Behavior**   The hard-regime hypothesis examines whether prediction errors and constraint violations concentrate in operational extremes, such as high-load conditions and near-capacity limits, and in structurally critical regions such as high-degree buses. Identifying these concentration patterns reveals where models are most brittle and where targeted architectural or training interventions may be required.

This reliability-centered perspective reflects broader challenges in scientific foundation modeling. Across domains such as fluid dynamics, molecular simulation, and climate science, learned models must respect conservation laws, symmetries, and geometric constraints, particularly under extreme conditions. While our empirical analysis focuses on ACOPF, the underlying relationship between data-driven approximation and hard physical feasibility is not unique to power systems. By studying ACOPF as a controlled and well-supervised constrained optimization problem, we aim to extract methodological insights that may inform foundation model design in other scientific settings where physical consistency and robustness under stress are critical.

## 3   FOUNDATION MODEL DESIGN PRINCIPLES

### 3.1   EXPERIMENTAL FRAMEWORK

To identify design principles for constrained scientific foundation models, we establish a controlled experimental framework spanning representative architectures, diverse training regimes, and systematic evaluation protocols across multiple network topologies.

**Model Architectures.**   We consider eight representative graph neural networks (GNN) backbones that are commonly used in prior work on learning-based OPF surrogates and cover both homogeneous (single node/edge type) and heterogeneous (typed nodes/edges) message passing. All models produce node-level predictions that are assembled into the ACOPF solution vector. In particular, we include three widely used homogeneous message-passing architectures that treat all nodes and edges uniformly: **GCN** Kipf (2016), **GAT** Veličković et al. (2017), and **GIN** Xu et al. (2018). We also include a **Graph Transformer** Yun et al. (2019) that applies multi-head attention over graph neighborhoods. For heterogeneous settings, we include four architectures: **RGAT** Busbridge et al. (2019), **HeteroGNN** Zhang et al. (2019), **HGT** Hu et al. (2020), and **HEAT** Mo et al. (2022). These methods use type-specific projections and/or relation-specific attention to enable different transformation and aggregation rules across node and edge types. We provide the explicit layer update equations used in our implementations in Appendix A.

**Dataset.** We base our experiments on OPFData Lovett et al. (2024), which provides solved ACOPF instances across ten representative power network topologies. For each topology, OPF-Data includes 300K feasible operating points obtained by perturbing load profiles and solving the corresponding ACOPF with a state-of-the-art solver. We use OPFData solutions as supervision and adopt fixed train/validation/test splits per topology to ensure consistent comparisons across models and objectives.

**Loss functions.** The baseline objective minimizes squared error (MSE) on solution variables. Constraint-aware objectives address this by incorporating constraint residuals directly into training. Augmented Lagrangian (AL) methods Bouchkati et al. (2024) incorporate constraints into training objectives, while violation-based Lagrangian (VBL) methods Fioretto et al. (2020) explicitly penalize constraint violations in the loss. We leave the detailed equations involved in these loss functions in Appendix B.

**Evaluation Metrics.** We evaluate all models using two complementary criteria: predictive accuracy and physical feasibility. Predictive accuracy is measured as the mean squared error (MSE) between predicted and ground-truth ACOPF solution variables. Physical feasibility is quantified by aggregating violations across each constraint family induced by the prediction. To enable fair comparison across network sizes, violation magnitudes are normalized by the square root of the topology size.

## 3.2 LEARNING ACROSS SYSTEMS VIA MULTI-TOPOLOGY PRETRAINING

Pretraining across heterogeneous systems enables models to learn reusable physical representations that transfer to unseen system structures, with fine-tuning providing system-specific calibration at a fraction of the training cost required when learning from scratch. A system in the ACOPF context refers to a network topology defined by transmission connectivity, generator placement, and load distribution. Across topologies, the number and types of components vary, as do specific constraint instances, yet the underlying physics remains invariant. Multi-topology pretraining learns these invariant physical laws from diverse network structures. Figure 1 compares architectures trained on single topologies (left panels) versus jointly on three topologies (right panels). Heterogeneous models (HGT, HEAT) maintain low violations under multi-topology training despite reduced per-topology samples, while homogeneous models (GCN, GAT, GIN) degrade substantially. Critically, although single-topology training appears superior on its training topology, it struggles under shift. Combining zero-shot transfer results from Table 1, we observe that Transformer, for example, achieves orders of magnitude improvement in solution quality and reduction in constraint violations when trained jointly on multiple topologies. This demonstrates that multi-topology pretraining's exposure to diverse structures outweighs the cost of reduced individual system coverage.

Table 1: Zero-shot transfer comparing Transformer and HGT architectures on single- and multi-topology pretraining. Best performance for each training→evaluation scenario shown in bold.

| | Trans.+MSE | | HGT+MSE | |
| --- | --- | --- | --- | --- |
| **Training → Eval** | **OPF Sol. Err.** | **Viol.** | **OPF Sol. Err.** | **Viol.** |
| case30 → case57 | **4.743** | 5.053 | 5.299 | **2.060** |
| case30 → case118 | **1.912** | 9.118 | 2.030 | **3.766** |
| case{57,118} → case30 | 1.120 | 4.853 | **0.4693** | **2.286** |
| case{30,118} → case57 | 5.677 | 1.990 | 5.6337 | **1.4781** |
| case{30,57} → case118 | **2.011** | 26.295 | 6.173 | **12.26** |

Additionally, multi-topology pretraining provides strong initialization for adaptation to new systems, as evidenced by fine-tuning efficiency on large target topologies. Figure 2 and Table 2 compare fine-tuning pretrained models against training from scratch on case118 and case500. Fine-tuning exhibits dramatically faster convergence, reaching equivalent feasibility thresholds in 48.8% fewer optimization steps on case118 (1.5M vs 3M samples) and 83.6% fewer steps on case500 (215K vs 1.31M samples).

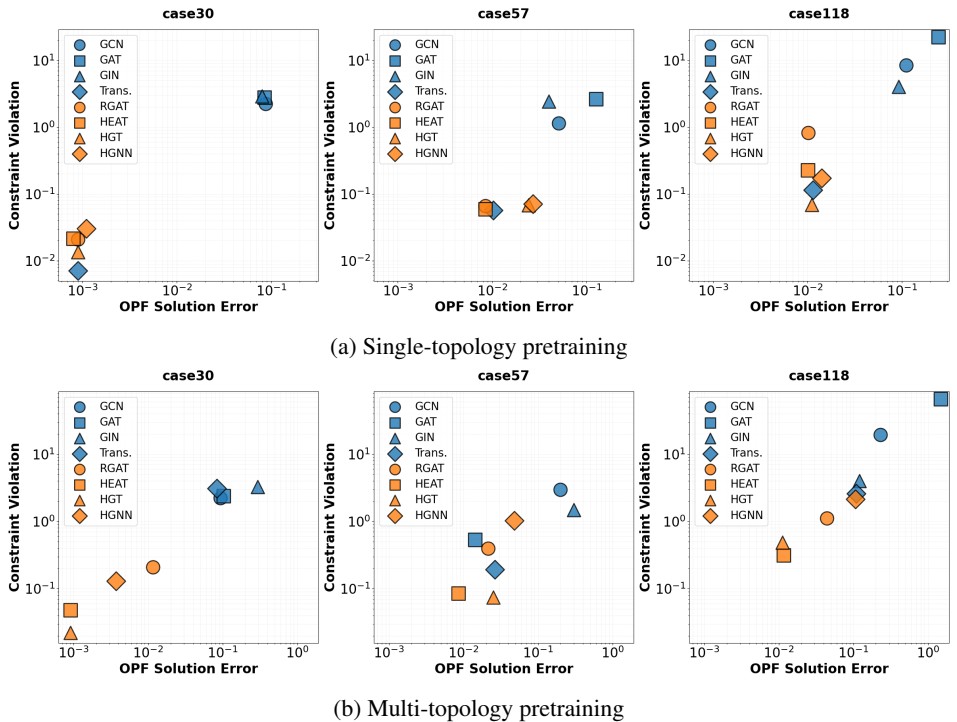

(a) Single-topology pretraining

(b) Multi-topology pretraining

Figure 1: Architecture comparison on single-topology (top) vs. multi-topology (bottom) pretraining. cases considered are case30, case57, and case118. Models trained with single topology are evaluated on the topology it is trained, while models trained jointly on multi-topology are evaluated on the each of the topology separately. Heterogeneous architectures generally demonstrate better performances than homogeneous models in both solution quality and constraint satisfaction, especially in multi-topology training.

Table 2: Convergence comparison between training from scratch and fine-tuning across grid sizes. Improvements in final violation and training steps are shown as blue subscripts.

| Case | Total Violation Norm | | Training Steps | |
|---|---|---|---|---|
| | Scratch | Finetuned | Scratch | Finetuned |
| 118 | 8.986 | $6.732_{(-25.1\%)}$ | 3,000,000 | $1,535,008_{(-48.8\%)}$ |
| 500 | 15.387 | $14.910_{(-3.1\%)}$ | 1,310,048 | $215,008_{(-83.6\%)}$ |

The convergence curves reveal that fine-tuned models rapidly suppress initial violations and stabilize early, while training from scratch exhibits prolonged high-variance dynamics and slower decay. Moreover, fine-tuning achieves superior final feasibility. This dual benefits, both faster convergence and better asymptotic performance, indicate that pretrained representations capture transferable physical structure that initializes models in a low-violation basin, accelerating optimization while improving constraint satisfaction.

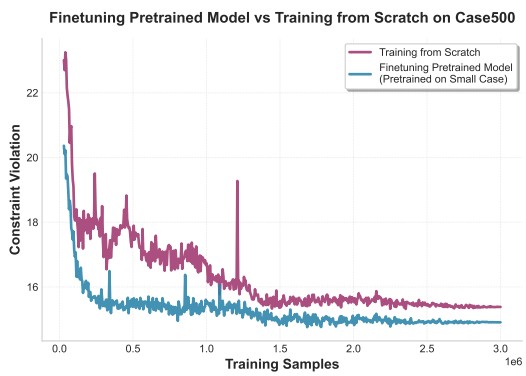

Figure 2: Constraint violation convergence on case500: fine-tuning vs. training from scratch

Overall, there are two important factors that enable the transfer of learned representations: (1) model architecture supporting cross-system generalization via explicit structural encoding (in heterogeneous models) or sufficient capacity (Transformer with global attention), (2) training exposure to diverse system. This principle extends to other scientific domains with structural heterogeneity, including computational fluid dynamics to pretrain across geometries to learn Navier-Stokes physics, and materials science to pretrain across crystal structures for transferable interaction potentials).

## 3.3 TRAINING EFFICIENCY AT SCALE

Mixed-precision training with BF16 (compared to full precision) accelerates optimization with gains that scale with problem size, reducing training time by 38.5% on case118 and 41.0% on case500. This scaling behavior indicates that BF16 primarily reduces compute and memory costs of large graph message passing and constraint evaluation rather than providing fixed speedup, moderating the steep runtime growth observed with FP32 as network size increases. These results establish mixed precision as a critical design consideration for foundation-style training on large systems, enabling substantial cost reductions without compromising model structure or feasibility-aware objectives.

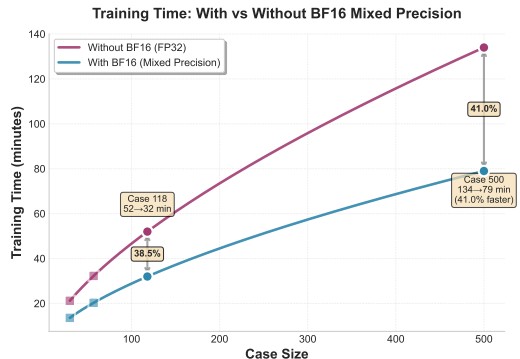

Figure 3: Training time vs. case size with and without mixed precision training (BF16)

## 3.4 CONSTRAINT-AWARE OBJECTIVES FOR ACTIONABLE RELIABILITY

When constraints encode scientific validity, training objectives must explicitly target feasibility, not only prediction fidelity. Standard supervised learning with MSE optimizes for prediction accuracy but treats constraints as implicit patterns to be learned through data. In constrained scientific systems, small prediction errors that appear acceptable in MSE can push solutions outside constraint boundaries, violating physical constraints that render predictions operationally unusable. MSE provides no mechanism to prioritize constraint satisfaction when accuracy and feasibility trade off.

Figure 4 demonstrates that architecture and loss function interact critically across training regimes. Transformer with MSE achieves low solution error on small single-topology tasks but degrades significantly with scale. Constraint violations increase an order of magnitude from case30 to case118. HGT with AL maintains consistent performance across all configurations. MSE produces the highest violations across all architectures, VBL offers moderate improvement, and AL achieves the strongest constraint satisfaction. Under distribution shift, constraint-aware objectives become essential: on zero-shot transfer comparing MSE and AL with HGT architecture shown in Table 3, AL achieves significantly smaller constraint violations, consistently regardless of the topology size while maintaining comparable solution accuracy. Linear probing analyses, presented in Figure 5 re-

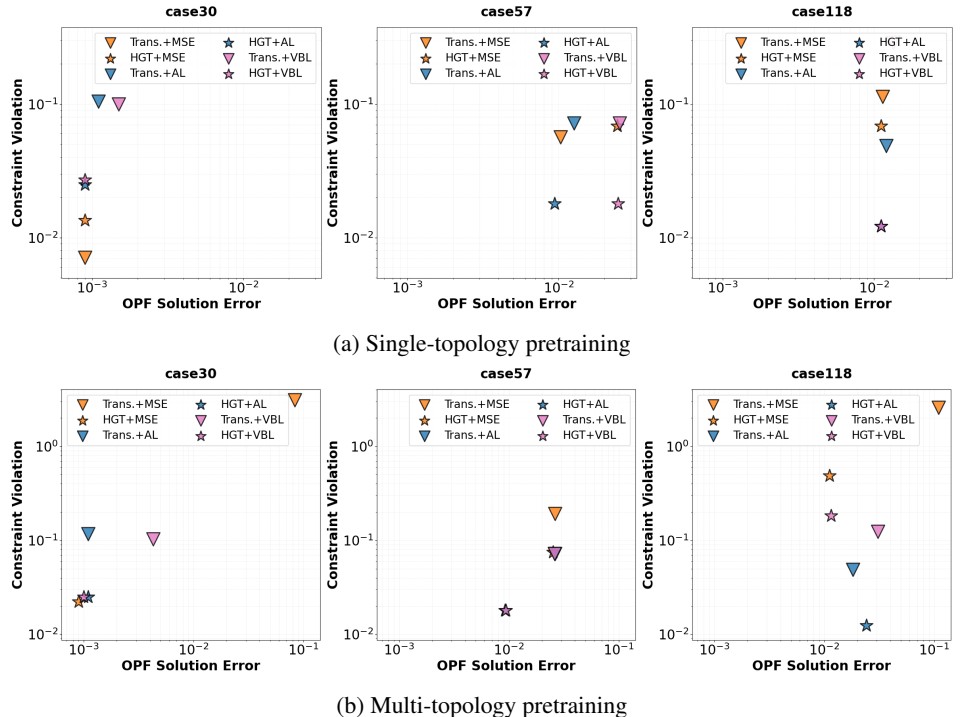

(a) Single-topology pretraining

(b) Multi-topology pretraining

Figure 4: Loss function comparison across two selected architectures (one homogeneous and one heterogeneous) under equal training budgets. Top panels: single-topology pretraining. Bottom panels: multi-topology training jointly on case30, case57. and case118. Each architecture is paired with three loss functions (MSE, AL, VBL). Models trained with single topology are evaluated on the topology it is trained, while models trained jointly on multi-topology are evaluated on the each of the topology separately. Constraint-aware loss functions, AL and VBL, achieve signficantly better constraint violations compared to MSE while AL performs slightly better than VBL.

veal that AL training induces increasingly nonlinear representations of physical grid characteristics across deeper layers, the same architecture produces markedly different activation structures under AL versus MSE for identical data suggesting that constraint-aware objectives shape learned geometry to encode physical structure beyond pattern matching, accounting for superior generalization to unseen topologies.

Table 3: Zero-shot transfer comparing MSE and AL on single- and multi-topology pretraining. Best performance for each training→evaluation scenario shown in bold.

| | HGT+MSE | | HGT+AL | |
|---|---|---|---|---|
| | **OPF Sol. Err.** | **Viol.** | **OPF Sol. Err.** | **Viol.** |
| case30 → case57 | 5.299 | 2.060 | 5.311 | **0.230** |
| case30 → case118 | 2.030 | 3.766 | 2.030 | **1.262** |
| case{57,118} → case30 | **0.4693** | 2.286 | 0.547 | **0.239** |
| case{30,118} → case57 | 5.6337 | 1.4781 | **5.225** | **0.018** |
| case{30,57} → case118 | 6.173 | 12.26 | 2.837 | **0.907** |

Constraint-aware objectives are essential whenever violations carry operational consequences. Among the objectives evaluated, AL consistently outperforms VBL for ACOPF, though practical deployment requires careful implementation: normalizing constraint residuals across physical units, using adaptive penalty schedules that increase constraint weighting as training progresses, and monitoring dual variable convergence. Even without hard feasibility guarantees, AL-trained surrogates remain practically valuable for scenario screening, contingency ranking, warm-starting conventional solvers, and workflows where approximate solutions accelerate computation without requiring strict feasibility at every step. Nonetheless, safety-critical dispatch decisions demand stronger guarantees,

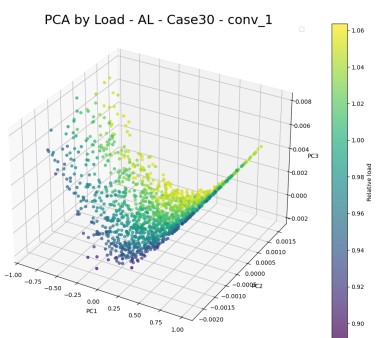 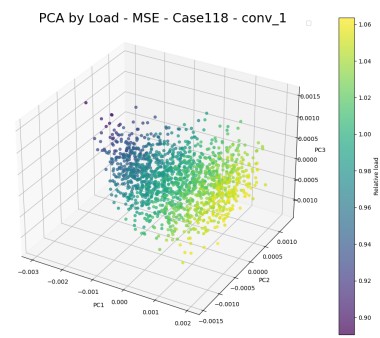

Figure 5: PCA components of activation for the top layer of convolutions in HGT trained on AL (left) vs MSE (right) losses. We see that both losses lead to the model capturing physical system load, but AL enforces a much more non-linear structure on the model's internal representation.

and future work should explore hard constraint enforcement with approaches including projection layers, certified feasibility methods, and hybrid solver-in-the-loop training.

## 3.5 RELIABILITY STRESS TESTS: EXTREMES AND STRUCTURAL HARD CASES

Average-case metrics hide scientific failure modes that emerge under operational extremes and structural vulnerabilities. Stress tests should target high-consequence regimes including peak loads, near-capacity conditions, topologically complex regions, where models are most brittle and conventional evaluation fails to expose risks that determine deployment viability. We identify two complementary failure modes through diagnostic analyses. First, we observe model errors and constraint violations concentrate at high-load operating points where constraint margins tighten as shown in Figure 6a. Homogeneous message-passing models (GCN, GAT, GIN) degrade sharply under load shift with noticeably weaker generalization to extreme conditions, while heterogeneous models (HGT, HEAT) and Transformers maintain more consistent performance. Second, we observe node-level error correlates with topological complexity measured by node degree ($r = 0.51$) irrespective of model type, illustrated in Figure 6b. Violations concentrate at high-degree buses that serve as network hubs, indicating that local structural complexity is a universal failure mode across architectures.

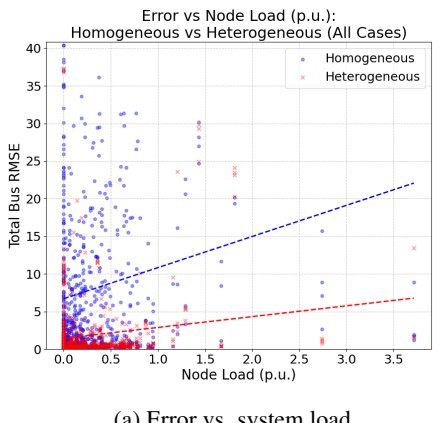 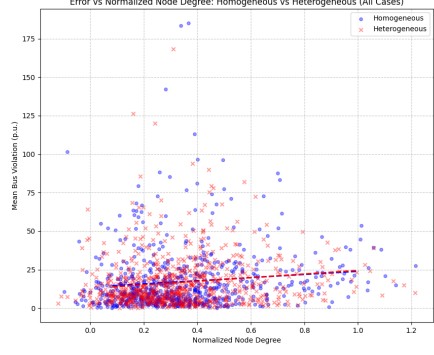

(a) Error vs. system load          (b) Error vs. topological complexity

Figure 6: Regime and structural stress tests reveal architecture-dependent failure modes. (a) Model error increases with system load across all architectures, but heterogeneous models (orange) generalize to high-load extremes significantly better than homogeneous models (blue). (b) Model error across categories with respect to node degree, up to case2000 with degree normalized with respect to case maximum degree. Model error by node correlates with node degree ($r = 0.51$).

These diagnostics inform operational workflows: (1) flag high-load scenarios and high-degree nodes for solver verification rather than relying on surrogate predictions, (2) use violation concentration patterns to guide targeted data augmentation: oversampling extreme regimes and structurally complex subgraphs during training, (3) implement topology-aware fallback strategies that route predictions through conventional solvers when operating near identified failure modes.

## 4 DISCUSSION: ROADMAP AND OPEN PROBLEMS FOR CONSTRAINED SCIENTIFIC FOUNDATION MODELS

Our empirical investigation of ACOPF surrogate learning across architectures, training objectives, and transfer scenarios yields actionable principles for building foundation models on constrained scientific systems. We distill these findings into a forward-looking roadmap that defines the frontier for constrained scientific foundation models.

**Representations and training strategies.** Heterogeneous architectures that explicitly encode component types and relational structure consistently outperform homogeneous baselines in maintaining feasibility under topology shift, though transformers with global attention can implicitly learn type separation given sufficient capacity. Multi-topology pretraining significantly reduces fine-tuning budgets on large systems, but optimal curriculum design that balances small diverse systems and large representative ones remains unclear. Constraint-aware Lagrangian objectives reduce violations by an order of magnitude but exhibit high hyperparameter sensitivity and longer pretraining time; in our experiments, all methods including AL were tuned with Bayesian HPO over penalty schedules and dual variable step sizes to ensure stable training. This motivates further exploration of adaptive penalty schedules and hybrid training that alternates between supervised and constraint-aware phases.

**Reliability under uncertainty and extremes.** Foundation models for high-consequence applications must provide calibrated uncertainty estimates that quantify feasibility risk—the probability of constraint violation under epistemic uncertainty. Standard neural uncertainty quantification (UQ) focuses on predictive variance rather than violation probabilities, requiring new methods that propagate uncertainty through constraint residuals and estimate tail risks efficiently. Prediction errors and constraint violations concentrate at high-load conditions and topologically complex nodes, revealing systematic brittleness in operational extremes where stakes are highest. Addressing this requires active learning strategies to oversample underrepresented extremes, adversarial data augmentation near constraint boundaries, and stress-test suites that probe worst-case scenarios beyond historical operating ranges. Evaluations for safety-critical deployments should correspondingly report high-percentile violation statistics (e.g., 95th and 99th percentile) stratified by operating regime, rather than relying solely on mean metrics, to surface worst-case behavior where operational consequences are highest.

**Deployment and operational integration.** Operational deployment requires hybrid architectures that use surrogates for rapid screening with solver verification on flagged cases, rollback strategies that detect infeasibility and fall back to conventional methods, and human-in-the-loop workflows that surface uncertainty to domain experts. Critical open questions include what metrics best indicate when models extrapolate beyond training distributions and how to design interfaces that communicate feasibility risk to non-ML-expert operators. Comprehensive evaluation protocols must report constraint-specific violation rates, tail statistics capturing worst-case behavior, and performance stratified by operating regime, with standardized zero-shot transfer tasks and ablation studies to enable reproducible research across scientific domains.

These open challenges define a research agenda for constrained scientific foundation models that integrates insights from machine learning, scientific computing, and domain-specific operational requirements. The LUMINA framework provides a starting point by demonstrating empirically grounded principles for architecture design, training objectives, and evaluation protocols on a representative constrained optimization problem, with open-source tools to support reproducible investigation across scientific domains.

## 5 CONCLUSION

We have extracted empirically grounded principles for building foundation models on constrained scientific systems through systematic investigation of AC optimal power flow surrogate learning across architectures, training regimes, and evaluation scenarios. Multi-topology pretraining yields transferable representations that enable zero-shot generalization and substantial reductions in fine-tuning budgets when adapting to new systems, demonstrating that foundation models can learn physics in a topology-agnostic manner while preserving the structural inductive biases necessary for cross-system transfer. Constraint-aware Lagrangian objectives materially improve feasibility under distribution shift relative to standard supervised learning, establishing explicit constraint penalization as essential for operational reliability beyond in-distribution accuracy. Diagnostic analyses reveal that violations concentrate at operational extremes and topologically complex nodes, defining stress-test protocols that probe model behavior where stakes are highest and conventional evaluation fails to expose brittleness. These principles suggest how to build scientific foundation models that accelerate decision-making in high-consequence applications, while respecting physical structure, maintaining constraint satisfaction, and providing interpretable failure modes that enable trustworthy deployment alongside conventional solvers. While empirical validation in domains beyond power systems remains future work, the principles identified, including heterogeneous structural encoding, multi-instance pretraining, and constraint-aware objectives, are domain-agnostic and directly applicable to any constrained scientific system with typed relational structure and governing conservation laws, such as computational fluid dynamics.

### ACKNOWLEDGMENTS

This material is based upon work supported by Laboratory Directed Research and Development (LDRD) funding from Argonne National Laboratory, provided by the Director, Office of Science, of the U.S. Department of Energy under contract DE-AC02-06CH11357.

An award of computer time was provided by the ASCR Leadership Computing Challenge (ALCC) program. This research used resources of the Argonne Leadership Computing Facility, which is a U.S. Department of Energy Office of Science User Facility operated under contract DE-AC02-06CH11357.

This research used resources of the National Energy Research Scientific Computing Center (NERSC), a Department of Energy User Facility using NERSC award ALCC-ERCAP0038201.

This work was supported by the National Research Foundation of Korea (NRF) funded by the Ministry of Science and ICT under Grant RS-2025-02215243.

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

The submitted manuscript has been created by UChicago Argonne, LLC, Operator of Argonne National Laboratory ("Argonne"). Argonne, a U.S. Department of Energy Office of Science laboratory, is operated under Contract No. DE-AC02-06CH11357. The U.S. Government retains for itself, and others acting on its behalf, a paid-up nonexclusive, irrevocable worldwide license in said article to reproduce, prepare derivative works, distribute copies to the public, and perform publicly and display publicly, by or on behalf of the Government. The Department of Energy will provide public access to these results of federally sponsored research in accordance with the DOE Public Access Plan (http://energy.gov/downloads/doe-public-access-plan).

## A  BASELINE MODEL UPDATE EQUATIONS

### A.1  HOMOGENEOUS MODELS.

Homogeneous GNNs use a shared set of parameters across all nodes and edges, and do not explicitly condition their message functions on node or edge types.

**GCN:** Graph convolutional networks Kipf (2016) perform normalized neighborhood aggregation. Using self-loops and $\tilde{A} = A + I$, $\tilde{D}_{ii} = \sum_j \tilde{A}_{ij}$, the layer update can be written as

$$h_i^{\ell+1} = \sigma \left( W^\ell \sum_{i,j \in N(i)} \frac{1}{\sqrt{d_i d_j}} h_j^\ell \right).$$

Note that GCNs only capture adjacency, ignoring edge features entirely.

**GAT:** Graph attention networks Veličković et al. (2017) learn attention weights over neighbors. For a single head,

$$\alpha_{ij}^\ell = \text{softmax}(\text{LeakyReLU}(a[Wh_i^\ell || Wh_j^\ell])),$$

with layer updates given by:

$$h_i^{\ell+1} = \sigma(\sum_{j \in \mathcal{N}} \alpha_{ij}^\ell Wh_j^\ell).$$

**GIN:** Graph Isomorphism networksXu et al. (2018) use an MLP after aggregating node features over a neighborhood like:

$$h_i^\ell = \text{MLP}^\ell((1 + \epsilon^\ell)h_i^\ell + \sum_{j \in \mathcal{N}(i)} h_j^\ell).$$

**Graph Transformer:** Graph transformers Yun et al. (2019) do not explicitly encode node or edge type information, but do have global attention, which may give a model a way to capture global node-node relationships that would otherwise be captured explicitly by node type. As in Yun et al. (2019) we model the query/key/value tuple for a given layer like:

$$Q^{(\ell)} = H^{(\ell)} W_Q^{(\ell)}, \quad K^{(\ell)} = H^{(\ell)} W_K^{(\ell)}, \quad V^{(\ell)} = H^{(\ell)} W_V^{(\ell)},$$

where $H^{(\ell)} \in \mathbb{R}^{|\mathcal{V}| \times d_\ell}$ stacks node embeddings. Attention is then computed as

$$\text{Attn}(Q, K, V) = \text{softmax}\left( \frac{QK^\top}{\sqrt{d_k}} + M \right) V,$$

where $M$ is an attention mask encoding adjacency-restricted attention.

## A.2 Heterogeneous Models.

Fully heterogeneous GNNs encode both node types $t_i$ and edge types $r_{ij}$, using type-specific transformations. This is well-matched to power-grid structure, where buses, generators, loads, and shunts play distinct physical roles and participate in different subsets of constraints.

**RGAT:** Relational graph attention network Busbridge et al. (2019) attempt to explicitly encode edge relations of different types by giving each edge category its own projection matrices

$$m_{ij}^\ell = W_{r_{ij}}^\ell h_j^\ell, \; \alpha_{ij}^\ell = softmax(score(h_i^\ell, h_j^\ell, r(i,j), e_{ij}))$$

where layer updates are then given by

$$h_i^\ell = \sigma(\sum_{j \in \mathcal{N}} (\alpha_{ij}^\ell m_{ij}^\ell)).$$

**HeteroGNN:** Heterogeneous graph neural networks Zhang et al. (2019), are a straightforward extension of GNNs that learn a feature projection for each node type and aggregate messages from node-type specific neighborhoods. Layer updates are given by:

$$h_i^{\ell+1} = \sigma(W_t^\ell h_i^\ell + \sum_{\tau \in \mathcal{T}} \text{AGG}(h_\tau^\ell)$$

where $\tau \in \mathcal{T}$ is a typed node from a neighborhood of adjacent nodes of that type.

**HGT:** Heterogeneous Graph Transformer (HGT) Hu et al. (2020) combines typed message passing with multi-head attention. For node $i$ attending to neighbor $j$ along edge relation $r_{ij}$ with node types $t_i, t_j$, HGT uses type-dependent projections giving Q/K/V like:

$$q_i^{(\ell)} = W_Q^{(\ell, t_i)} h_i^{(\ell)}, k_j^{(\ell)} = W_K^{(\ell, t_j)} h_j^{(\ell)}, v_j^{(\ell)} = W_V^{(\ell, t_j)} h_j^{(\ell)}$$

An attention model that attends to type-specific connections like:

$$s_{ij}^\ell = \frac{\left(q_i^{(\ell)}\right)^\top \left(W_A^{(\ell, r_{ij})} k_j^\ell\right)}{\sqrt{d_k}}, \; \alpha_{ij}^{(\ell)} = \text{softmax}_{j \in N(i)}(s_{ij}^\ell).$$

**HEAT:** Heterogeneous edge attention transformers Mo et al. (2022) are a category of transformer that augments typed attention with additional edge-enhancement. HEAT follows the same typed message passing template as above, with attention scores and messages explicitly conditioned on $(t_i, t_j, r_{ij}, e_{ij})$:

$$m_{ij}^{(\ell)} = \phi^{(\ell)}\left(h_i^{(\ell)}, h_j^{(\ell)}, e_{ij}, t_i, t_j, r_{ij}\right), \; h_i^{(\ell+1)} = \sigma\left( \sum_{j \in N(i)} \alpha_{ij}^{(\ell)} m_{ij}^{(\ell)} \right),$$

where $\phi^{(\ell)}(\cdot)$ is a learnable message function and $\alpha_{ij}^{(\ell)}$ is computed by a typed attention mechanism.

# B  LOSS FUNCTIONS

**Pointwise regression (MSE).** MSE objective is defined by

$$L_{\text{MSE}}(\theta) = \mathbb{E}\left[\|\hat{\mathbf{y}} - \mathbf{y}\|_2^2\right].$$

This objective measures predictive accuracy but does not explicitly enforce feasibility.

**Augmented Lagrangian (AL).** To encourage feasibility during training, we incorporate constraint residuals using an augmented Lagrangian objective Bouchkati et al. (2024) as follows:

$$
\begin{aligned}
L_{\text{AL}}(\theta; \boldsymbol{\lambda}, \boldsymbol{\mu}, \rho) = L_{\text{MSE}}(\theta) &+ \boldsymbol{\lambda}^T \mathbf{r}(\hat{\mathbf{y}}) + \frac{\rho}{2}\|\mathbf{r}(\hat{\mathbf{y}})\|_2^2 \\
&+ \boldsymbol{\mu}^T \max\{\mathbf{h}(\hat{\mathbf{y}}), 0\} + \frac{\rho}{2}\|\mathbf{h}(\hat{\mathbf{y}})\|_2^2,
\end{aligned}
$$

where $\boldsymbol{\lambda}$ and $\boldsymbol{\mu}$ are dual variables associated with equality constraints and inequality constraints, respectively, and $\rho > 0$ is a penalty parameter. During the training for $\theta$, $(\boldsymbol{\lambda}, \boldsymbol{\mu})$ are updated periodically using ascent steps with projection for nonnegativity:

$$\boldsymbol{\lambda} \leftarrow \boldsymbol{\lambda} + \rho\,\mathbf{r}(\hat{\mathbf{y}}), \qquad \boldsymbol{\mu} \leftarrow \boldsymbol{\mu} + \rho\,\max\{\mathbf{h}(\hat{\mathbf{y}}), 0\}.$$

This objective adaptively reweights constraint satisfaction during training and typically reduces violations under distribution shift.

**Violation-based Lagrangian (VBL).** As a feasibility-aware alternative, we use violation-based Lagrangian objective Fioretto et al. (2020) as follows:

$$L_{\text{VBL}}(\theta; \boldsymbol{\lambda}, \boldsymbol{\mu}) = L_{\text{MSE}}(\theta) + \boldsymbol{\lambda}^T |\mathbf{r}(\hat{\mathbf{y}})| + \boldsymbol{\mu}^T \max\{\mathbf{h}(\hat{\mathbf{y}}), 0\},$$

where $\boldsymbol{\lambda}, \boldsymbol{\mu} \geq 0$ are the Lagrangian multipliers that control the accuracy-feasibility trade-off. Similar to AL, the dual variables are updated periodically using ascent steps with step size $\rho > 0$:

$$\boldsymbol{\lambda} \leftarrow \boldsymbol{\lambda} + \rho\,|\mathbf{r}(\hat{\mathbf{y}})|, \qquad \boldsymbol{\mu} \leftarrow \boldsymbol{\mu} + \rho\,\max\{\mathbf{h}(\hat{\mathbf{y}}), 0\}.$$

This objective adaptively weights constraint violation degrees, as compared to the satisfiability degree Fioretto et al. (2020).

