# OpenReview forum: "LUMINA: Foundation Models for Topology Transferable ACOPF"
_ICLR.cc/2026/Workshop/FM4Science — ICLR 2026 Workshop FM4Science Poster_

### Official Review · Reviewer_w8Uy · 2026-02-20
**This paper introduces LUMINA, a framework for investigating design principles for scientific foundation models through the AC Optimal Power Flow problem.**

**Rating:** 5
**Confidence:** 3

**Review:**

Pros:
Extensive Empirical Breadth: The study rigorously compares homogeneous and heterogeneous backbones, demonstrating that explicit structural encoding is vital for topology-agnostic generalization. Significant Pretraining Efficiency: Multi-topology pretraining significantly accelerates adaptation, reducing training steps on large systems by 83.6% compared to training from scratch. Actionable Reliability Metrics: The paper identifies that errors concentrate at high-load regimes and high-degree buses, providing a roadmap for targeted solver fallback strategies. Feasibility Gains: Using Augmented Lagrangian (AL) objectives reduces constraint violations by an order of magnitude over standard MSE by inducing physics-aware internal representations.

Cons:
Incomplete Feasibility Guarantee: While AL materially improves reliability, it lacks the hard constraint enforcement required for high-consequence grid operations.
Hyperparameter Sensitivity: The authors acknowledge that Lagrangian-based pretraining is sensitive to penalty schedules and hyperparameters, complicating broad deployment.
Limited Domain Validation: The foundation model claims are based solely on ACOPF; empirical proof of transferability to other scientific domains (e.g., fluid dynamics) is missing.

---

### Official Review · Reviewer_CM6w · 2026-02-23
**Solid Empirical Study on ACOPF Models**

**Rating:** 7
**Confidence:** 3

**Review:**

This paper presents LUMINA, a framework for studying topology-transferable surrogate models for AC optimal power flow (ACOPF) under hard physical constraints. The authors conduct a comprehensive empirical comparison of multiple graph-based architectures and training objectives, including standard supervised learning and constraint-aware Lagrangian formulations. Experiments span single- and multi-topology pretraining, zero-shot transfer, fine-tuning efficiency, and stress tests under high-load and structurally complex regimes. Based on these results, the paper distills several practical design principles for improving feasibility and reliability in constrained scientific learning settings.

Pros
1. Addresses a practically important problem where constraint satisfaction is essential for real-world deployment.
2. Provides a broad and careful empirical comparison across architectures, training objectives, and transfer settings.
3. Multi-topology pretraining and stress-test analyses offer useful insights into when models fail (e.g., high-load and high-degree nodes).
4. The paper is clearly structured, and the experimental pipeline appears reproducible.

Cons
1. While the paper adopts the term “foundation model,” the empirical study remains confined to a single scientific task (ACOPF) and one dataset family. The contribution is therefore best understood as multi-topology pretraining within a specific constrained optimization problem.
2. Since feasibility and reliability are central to the paper’s motivation, it would strengthen the evaluation to include worst-case or high-percentile violation statistics, rather than only averaged violation metrics. In high-risk systems like power grids, even rare constraint violations can be critical.

---

### Decision · Program_Chairs · 2026-03-02

Accept (Poster)